# Examining the effect of non-specialised clinical rotations upon medical students' Thanatophobia and Self-efficacy in Palliative Care: a prospective observational study in two medical schools

Guilherme Gryschek,[1] Dario Cecilio-Fernandes [ID] ,[2] Guilherme Antonio Moreira de Barros,[3] Stephen Mason,[4] Marco Antonio de Carvalho-Filho [ID] [5,6]

For numbered affiliations see end of article.

**Correspondence to**
Dr Marco Antonio de Carvalho-Filho;
m.a.de.carvalho.filho@umcg.nl

## ABSTRACT

**Introduction** Including palliative care (PC) in overloaded medical curricula is a challenge, especially where there is a lack of PC specialists. We hypothesised that non-specialised rotations could provide meaningful PC learning when there are enough clinical experiences, with adequate feedback.

**Objective** Observe the effects of including PC topics in non-specialised placements for undergraduate medical students in two different medical schools.

**Design** Observational prospective study.

**Setting** Medical schools in Brazil.

**Participants** 134 sixth-year medical students of two medical schools.

**Methods** This was a longitudinal study that observed the development of Self-efficacy in Palliative Care (SEPC) and Thanatophobia (TS) in sixth-year medical students in different non-specialised clinical rotations in two Brazilian medical schools (MS1 and MS2). We enrolled 78 students in MS1 during the Emergency and Critical Care rotation and 56 students in MS2 during the rotation in Anaesthesiology. Both schools provide PC discussions with different learning environment and approaches.

**Primary outcomes** SEPC and TS Scales were used to assess students at the beginning and the end of the rotations.

**Results** In both schools' students had an increase in SEPC and a decrease in TS scores.

**Conclusion** Non-specialised rotations that consider PC competencies as core aspects of being a doctor can be effective to develop SEPC and decrease TS levels.

## INTRODUCTION

Medicine has made significant advances in improving healthcare outcomes during the last century. While life expectancy has increased in many countries, chronic and degenerative diseases have become the leading causes of death.[1][2] To deal with this new epidemiological context, doctors need to become competent to deliver palliative and end of life care.[2–4] To prepare doctors to cope with this new reality of practice, medical educators have the responsibility to develop pedagogical strategies to guide medical students towards becoming competent in supporting the psychological, social and spiritual needs of dying patients and their families.

The social cognitive theory suggests that individuals learn and change behaviour through intra-personal motivational constructs, known as 'self-efficacy' and 'outcome expectancy'. 'Self-efficacy' relates to the confidence an individual (in this case, medical students) has in their capacity in performing a specific task/behaviour. Students' knowledge, skills and previous experience, including the observation of other's performance, modulates their self-efficacy.[5] 'Outcome expectancy' refers to the perceived consequence of the target behaviour, together with the aggregate

### Strengths and limitations of this study

► This study access students' Self-efficacy in Palliative Care of two Brazilian medical schools, where palliative care is still beginning.
► Research took place in medical schools that do not have specialised palliative care placement.
► Observational study could determine if palliative care teaching inserted in non-specialised placements have effects on students' Thanatophobia and Self-efficacy in Palliative Care.
► Investigate this topic could bring new evidence on palliative care teaching for medical students in countries with less developed palliative care.

BMJ

value this behaviour brings to the student.[5 6] A 'positive' performance is more likely when students' self-efficacy is higher, and the expected outcome is rewarding.[5] Thus, appropriate training should not only increase students' knowledge, but also strengthen their self-confidence in achieving their objectives (self-efficacy), while showing the value of the desirable actions (outcome expectancies).[5 7] Social cognitive theory has been employed in studies to assess educational interventions, particularly where practical assessment of the development of knowledge, skills and attitudes are difficult to apply efficiently at scale.[8 9] Consequently, based on this theory, evaluating the design of learning activities should incorporate the measurement of self-efficacy and outcome expectancy.

The Self-efficacy in Palliative Care (SEPC) and the Thanatophobia (TS) scales were developed to evaluate student's self-efficacy and outcome expectancies in palliative care (PC).[6 7 10] The SEPC covers core aspects of doctors' expected behaviours in PC: effective communication with the patient and family; assessment and management of patient's symptoms and needs and multidisciplinary work within a healthcare team.[6] TS (literally 'fear of death') relates to healthcare professionals' avoidance of terminally ill patients.[11] Lower levels of TS are considered desirable when healthcare professionals are providing end-of-life care; therefore, the TS scale may be used to measure the outcomes expectancies related to PC education.[6 11] Both scales (SEPC and TS) were validated to measure the impact of medical training in PC learning.[6]

Although medical schools are recommended to include PC in their curricula, the teaching and assessment strategies are heterogenous and many schools do not have access to PC specialised services for students' placements.[12 13] Examples of specialised services are hospices, wards and outpatients' clinics run by palliative medicine specialists. While in Europe and other countries the PC have wider offer and students training are based in specialised services, in Latin America that is not the case.[14 15] For example, in Brazil, there are less than 200 specialised PC services, and many are not linked to medical schools.[16] In practical terms, most medical students in Brazil do not have access to specialised training settings, which is the case of the medical schools in our research.[17] In Brazilian medical school, the overloaded curricula and the lack of specialised PC teachers hinder PC undergraduate education.[13] Therefore, we explore in this article whether a focus on PC topics during non-specialised clinical rotations without the active participation of PC specialists may offer efficient learning opportunities for senior medical students and bridge a gap within existing curricula.

Engagement and learning within a specialised practice may be considered ideal for training purposes. However, effective learning is mostly related to how any given scenario is modulated by students and supervisors to optimise the learning process through different lenses.[18 19] Among these lenses, the Kolb's cycle offers a compelling insight into understanding the learning process triggered by practical experiences.[19] Kolb's cycle stresses the importance of integrating the practical experience with the abstract conceptualisation of this experience. Therefore, the learning process occurs through consecutive cycles of lived experiences, reflexive observations and active planning of the next experimentations supported by appropriate supervision and feedback, in which students integrate, modulate and potentialise their own learning process.[18] Thus, we hypothesise that non-specialised placements can provide a proper learning environment to offer meaningful and productive clinical experiences, which could be capable of grounding the improvement of SEPC and the decreasing of TS levels.[19]

To verify our hypothesis, we assessed PC training outcomes in two different medical schools in Brazil. The first medical school offers a clinical rotation with direct contact with critical care and dying patients and opportunities to reflect and discuss the care for dying patients are inserted during the placement. The clinical supervisors within this first school (MS1), although not specialised in PC, accept end-of-life care as part of their professional and educational role as general internal medicine doctors. The second medical school (MS2) offers a shorter placement in anaesthesiology rotation which includes an outpatient service for PC and chronic pain patients and a homecare service to vulnerable patients who have a low performance-status. In the MS2 context, medical students have the opportunity to discuss PC topics, such as pain and symptoms management. In this longitudinal study, we aimed to measure medical students' SEPC and TS levels, before and after those two different rotations in the aforementioned medical schools. Our hypothesis is that non-specialised rotations are capable of nurturing SEPC and decreasing TS levels in undergraduate medical students.

## METHODS
### Context
The undergraduate medical course in Brazil is 6 years long. National guidelines for curriculum development demand the integration of learning activities with healthcare services throughout the training. In general, during the first 2 years, students attend classroom-based teaching, with a combination of lectures and small group discussions. Besides the classroom activities, students have regular placements in primary care and hospital facilities to be introduced to the logic of patient care. During years 3 and 4, students are trained in history taking, physical examination and clinical reasoning and they are involved in the longitudinal care of patients in primary care facilities. During the last 2 years, students undertake an internship, where they rotate in diverse clinical placements, such as internal medicine, paediatrics, primary care, gynaecology, surgery, critical care and medical emergencies. In these clinical rotations, students work under supervision and take direct responsibility for patient care in clinical and surgical wards, outpatient clinics and

emergency departments. After concluding their undergraduate studies, they receive full registration to practice medicine and are allowed to perform independently, even without resident training. Many new doctors start working at emergency departments and primary care facilities, despite their lack of postgraduate training.

## Design, setting and participants

We conducted a longitudinal observational study with two cohorts of year 6 medical students, from two separate medical schools, in 2017. Both medical schools are ranked in the top 10 within Brazil.[20 21] Below we briefly describe some aspects of the rotations and details of MS1 and MS2 placements are in online supplemental appendix section.

*MS1*: Final year students undertake an 8-week mandatory emergency and critical care placement, with 14 students per group. The first group took part in the research during the beginning of the final year, in January 2017, and the last one in September 2017. During this rotation, students engage in clinical activities for 11 hours daily, under the direct supervision of clinical teachers and experience the care of chronically ill and dying patients. It is hard to quantify exactly how many dying patients the students have contact during the rotation. However, as the rotation takes place in a non-surgical intensive care unit (ICU), many patients have chronic diseases. such as chronic obstructive pulmonary disease, heart failure and metastatic cancer. This ICU has 28 beds and a mortality rate of 30%. Unfortunately, in our context is still common that terminal patients (even those that should be considered for exclusive PC) receive their end-of-life care in the ICU or emergency room, as we lack PC services in the health network. Considering that, most of MS1 students are involved in caring for patients and assist them at the end of life and almost all testify the deaths of some of the patients during the rotation period. The clinical supervisors decided to include flexible PC discussions in bedside rounds due to the opportunities provided by this placement. Conversely, the supervisors provide formally structured discussions about end-of-life ethics in multi-professional meetings. Also, students have access to training sessions with simulated patients to develop communication skills (including breaking bad news) and emotional competence.[22 23] As such, students have several opportunities to acquire the knowledge, skills and attitudes required to care for dying patients, and reflexive learning is stimulated throughout the rotation by the supervisors. Nevertheless, most of the discussions during the rotation are about emergency medicine and critical care, and the activities around the topic of PC are complementary.

*MS2*: Final year medical students undertake a 6-week mandatory rotation in anaesthesiology and pain therapy, in groups of 12 students. The first group answered to this research in February 2017 and the last one in September 2017. Significant part of the training in this rotation is dedicated to anaesthesia abilities training. Within this period, students also have 32 hours dedicated to specific PC topics and experiences, spread in 3 weeks. Learning opportunities include lectures on PC concepts and pain management, home care visits for PC patients and supervised activities in the outpatient clinic under the supervision of PC consultants. In this placement, students work together with a multi-professional team, which includes social workers, psychologists, nurses, physical therapists and nutritionists. The students learn how to manage symptoms, work in a multidisciplinary team and deliver care for PC patients in different settings, such as outpatient clinics and home care. Aside from PC topics, students also engage in learning activities on anaesthesiology.

## Procedure

Data were collected from February 2017 to September 2017, when we send the invitation for participation in the study to all the students enrolled in the aforementioned rotations at MS1 and MS2 (80 and 60 students, respectively). Two authors, GG and GAMdB (supervisor in MS2), personally invited the students during the introductory activities of each rotation in MS1 and MS2, respectively. Both explained to students the research goals, the nature of the instruments that would be applied and guaranteed the anonymity of the participants. Authors also assured that participating in the research would not affect students' grades in that rotation, trying to minimise students' perception of being evaluated and its interference in data provided. Students who agreed to participate signed a written informed consent. We collected data from each group of students (MS1 and MS2) in two different time-points of their placements: the first point was at the start of the placement; the second one was during the last week. To report this study we used Strengthening the Reporting of Observational Studies in Epidemiology cohort reporting guidelines.[24]

## Questionnaires

*Demographic survey*: The authors developed a demographic questionnaire to explore students' previous personal and professional experiences related to PC and end-of-life care (online supplemental appendix), as it was expected that past experiences would influence students' learning and performance.[25 26]

*SEPC*[6]: The SEPC measures the self-perceived efficacy in PC by recording students' ratings of their confidence in performing PC-related actions, using a 100 mm Visual Analogue Scale. The Visual Analogue Scales range from 'would feel very anxious' to 'would feel very confident'. The point assigned on the Visual Analogue Scale is measured and scored from 0 up to 100. The 23-item SEPC has three subscales of (a) communication (8 items; score ranging from 0 to 800); (b) patient management (8 items; score ranging from 0 to 800); (c) multi-disciplinary team working (7 items; score ranging from 0 to 700).

*TS Scale*[11]: The original TS has 7 items and assesses attitudes related to the care for dying patients. The scale offers a reliable measure of students' outcome expectancy related to PC.[6] Participants rate how much they agree/

disagree with expressed negative attitudinal statements, with the final scores ranging from 7 to 49. Higher ratings indicate higher TS levels.

Both original scale was translated and validated to Brazilian Portuguese before this study.[27]

### Patient and public involvement
We did not involve patients directly in this study.

### Data analysis
Continuous variables were described by mean, minimal and maximal values and categorical variables as absolute (n) and relative frequency (%). We analysed the difference between students' premain and postmain scores on the scales using t-tests. Effect size was calculated by Cohen's d and used for checking the amplitude of changes in pretest and post-test scores, providing an index to assess the magnitude of the difference detected on t-test results. The effect size was characterised as small, medium, large or very large and evaluated the magnitude of change despite the sample size.[28] Bonferroni was used to correct the α value.[29] Therefore, we used α value of 0.0125 for all analyses. Data were analysed using IBM-SPSS V.21.0.[30]

### RESULTS
From the total students invited, two students from each school were excluded due to incomplete data. Thus, we included 78 students from MS1 and 56 students in MS2 (65.0% and 62.2% of last year students, respectively). Students' demographic data and previous experiences are summarised in table 1, which were answered in pretest data collecting.

In both schools, students had a significant increase in self-efficacy in all dimensions of the SEPC (p<0.001 for all analyses). Students' TS levels significantly decreased in MS1, but not in MS2, as shown in table 2 (significant results are highlighted). The effect size of the change in SEPC was larger at MS1 when compared with MS2. The effect size magnitude of TS was moderate in MS1.

### DISCUSSION
In this study, the SEPC and TS measured significant improvements in students' SEPC and TS levels, after the placements analysed, except in MS2, in which TS changes were not significant. These results support that non-specialised rotation can provide PC training with effects in students' SEPC and TS. Most of the students in both schools informed some previous experience with dying people, or as a personal experience or during their professional training. This could suggest that previous clinical placements give opportunities for caring of dying patients. Nevertheless, the placements analysed in this study gave to students an additional opportunity to care of dying patients and purposely learn and discuss PC topics.

In both schools, the placements brought improvements in students' SEPC, in all subscales, as expected after a learning experience that includes PC topics and discussions. Another study with UK medical students had similar results.[8] Nevertheless, in the UK study, the effect size magnitude was larger in almost every subscale of SEPC, although the pretest results were similar. One explanation for this could be the specialised PC placement the UK students had. There are other factors to be considered: time dedicated to PC training, specialised supervisors and services, placement length and the

| **Table 1** Descriptive data of MS1 and MS2 | | |
|---|---|---|
| | **Medical school 1** | **Medical school 2** |
| Sample size | 78 | 56 |
| Average age in years (min–max) | 24.8 (22–30) | 25.2 (22–34) |
| *Gender* | | |
| Male | 36 (46.2%) | 30 (53.6%) |
| Female | 42 (53.8%) | 26 (46.4%) |
| *Has anybody that you have been close to died?* | | |
| Yes | 63 (80.8%) | 45 (80.4%) |
| No | 15 (19.2%) | 11 (19.6%) |
| *Have you ever witnessed the moment of a person's death?* | | |
| Yes | 64 (82.1%) | 52 (92.9%) |
| No | 14 (17.9%) | 4 (7.1%) |
| *During your professional training, have you ever taken care of anybody that eventually died?* | | |
| Yes | 75 (96.2%) | 55 (98.2%) |
| No | 3 (3.8%) | 1 (1.8%) |

MS1 and 2, medical schools.

**Table 2** SEPC and Thanatophobia score means, t-test and effect size at medical schools 1 and 2

| Instruments | Pretest | Post-test | Difference post-pre | t | P value | 95%CI | Cohen's d | Effect size magnitude |
|---|---|---|---|---|---|---|---|---|
| *Medical school 1 (n=78)* | | | | | | | | |
| SEPC—communication | 346.50 (145.17) | 573.90 (124.47) | 227.39 (138.50) | 14.50 | <0.001 | 258.62 to196.17 | 1.64 | Very large |
| SEPC—patient management | 338.38 (112.56) | 527.50 (113.55) | 189.11 (140.30) | 11.90 | <0.001 | 220.75 to157.48 | 1.35 | Very large |
| SEPC—multidisciplinary teamwork | 361.26 (124.28) | 470.35 (105.77) | 109.09 (118.86) | 8.11 | <0.001 | 135.17 to81.01 | 0.92 | Large |
| TS | 21.54 (7.56) | 16.54 (7.58) | −5.00 (5.91) | −7.06 | <0.001 | 3.59 to6.41 | 0.79 | Moderate |
| *Medical school 2 (n=56)* | | | | | | | | |
| SEPC—communication | 296.41 (147.41) | 398.00 (171.64) | 101.59 (151.30) | 5.02 | <0.001 | 142.11 to61.07 | 0.67 | Moderate |
| SEPC—patient management | 327.86 (133.91) | 462.07 (126.55) | 134.21 (132.69) | 7.57 | <0.001 | 169.75 to98.68 | 1.01 | Large |
| SEPC—multidisciplinary teamwork | 388.50 (120.18) | 454.96 (120.49) | 66.46 (136.82) | 3.63 | 0.001 | 103.10 to29.83 | 0.49 | Small |
| TS | 22.82 (8.77) | 21.20 (7.67) | −1.62 (5.45) | 1.96 | 0.056 | −0.14 to 3.07 | 0.26 | Small |

PC, palliative care; SEPC, Self-efficacy in Palliative Care; TS, thanatophobia.

learning environment. In this case, a specialised placement, focused on training PC competencies, could bring development to students' self-efficacy and decrease their TS. In MS1, a decreasing in TS was observed, but not in MS2. Maybe this is a consequence of the experience in MS1 during the placement is related to more experience with critical and dying patients (see online supplemental appendix).

Worldwide, medical educators have devised different strategies for PC training and there is no 'gold standard' to guide best practices.[12 31 32] Although training in specialised PC services could bring larger improvement in SEPC, these services are not common in Brazil and are not an option for training medical students. Including a PC specialised placement in the already busy and crowded medical curriculum poses a challenge for medical educators. The challenge has two edges; one is the scarcity of specialised services and professionals to facilitate training, particularly in the developing world (where the PC is still incipient and needs substantial development); the second is the lack of space in the curriculum.

Ultimately, clinicians in most specialties, such as internal medicine, geriatrics, oncology, emergency medicine, ICU, primary care, paediatrics and so on, will have PC patients among their caseloads. Although there can often be a lack of PC services, different disciplines and placements should include PC in their curricula and provide PC training for all medical students.[33] Teachers could customise different clinical placements to modulate learning possibilities to students considering the resources available as shown in this study medical schools. By doing so, medical educators could ensure a psychological safety environment for learning, including feedback routines and the development of self-efficacy, which we considered as a fundamental aspect in schools we analysed.[34] Our study showed that non-specialised rotations, which include specific topics in PC, were capable of improving students' SEPC and decreasing their TS levels.

Therefore, others aspects as learning environment and experience, feedback opportunities and the inclusion of PC as part of clinical caring, could play relevant elements for PC training in this context.[19] We believe that this finding is related to opportunities to reflect on real end-of-life cases that were provided for the students during these placements. As anticipated by Kolb's experiential learning theory, the several occasions to reflect in and on the practice could have nurtured confidence and a readiness to practice PC.[19] In fact, the social cognitive theory states that self-efficacy and outcome expectancy are both modulated by the feedback (perceived support) from supervisors, which will eventually, impact on the internalisation of the desired behaviour.[5] Possibly offering opportunities to experience and reflect, guided by qualified supervision with structured feedback, may be as important as having access to specialised PC centres for the learning of undergraduate students.[5 7]

The challenge posed to medical educators is building up a meaningful learning process in complex workplaces

in different clinical rotations, despite the lack of ideal circumstances.[18] Clinical teachers need to nurture the learning experience of students embracing the contextual elements, even the negative ones, as teaching assets and not as pitfalls. Students will practice medicine in the real world and need to learn how to deal with the context of the workplace. Those with higher self-efficacy are more likely to reflect on their challenges and experiences to improve performance.[5] In this sense, reflection is essential not only to learn but also to feel the urge to change and improve the healthcare delivered to patients.[9] Therefore, including structured PC discussions in clinical rotations may offer an opportunity for teachers not only to prepare students to provide proper PC but also to open the debate about the relevance of including PC into their future practice.

### Strengths and limitations

This study had a consistent methodological approach and involved two different medical schools, with different undergraduate curricula, which included PC training in non-specialised rotations. We demonstrated that, although the two groups of students had different approaches in the sense of learning PC, both groups had improvements in their self-efficacy, even when students did not have access to specialised PC services.

Nevertheless, we should acknowledge some limitations in our study. First, we did not access students' conceptual knowledge of PC, neither the effects of other factors as placement length nor learning approaches. In this sense, as we did not control other factors, we cannot conclude if the changes in scales are related to daily placements' clinical situations or to specific teaching approaches. An interventional study with control groups would be more appropriate to verify if and how the different educational approaches could have different impacts on PC and TS levels.

Second, although the students have expressed more confidence in their performance, we cannot conclude that the improvements in SEPC and TS, observed in this study, will be persistent for the actual medical practice of the students. The proper assessment for this will require further evaluation over the new doctors in their practice.

Also, improvements in self-efficacy, in general, can be related to the opportunity given to medical students to reflect on their clinical experiences under the guidance of motivated facilitators.[5] This general improvement could have also been extrapolated to PC competencies.[7] Additionally, in MS2, as one of the researchers was supervisor, a socially desirable answer from the participants cannot be excluded.

Furthermore, the findings in this study are limited to two Brazilian medical schools and any generalisation should be cautious. Future studies are needed to verify whether our findings will be consistent across different contexts.

### Implications for practice and next steps

In a context of an overloaded curriculum and a lack of departments and services of palliative medicine in medical schools, our findings suggest that inserting PC topics in different non-specialised rotations could benefit the development of medical students SEPC and reduce their TS. Thus, a vertical insertion of PC concepts and practices along the course, in different disciplines and rotations, could address PC learning needs for undergraduate students, even without specialised PC services for medical training.

We still need to understand the impact of implementing PC training on clinical rotations on students' declarative and procedural knowledge acquisition. It is also essential to clarify what are the teaching methodologies related to better development of PC competencies in undergraduate medical students.

## CONCLUSION

Our findings support that non-specialised rotations can include PC basic competencies as part of the training. Medical schools could prepare future doctors to provide PC in any area it is required by embracing PC as a core competence, indispensable for becoming a doctor.

**Author affiliations**
[1] Internal Medicine, University of Campinas School of Medical Sciences, Campinas, SP, Brazil
[2] Department of Medical Psychology and Psychiatry, University of Campinas School of Medical Sciences, Campinas, SP, Brazil
[3] Department of Anesthesiology, Universidade Estadual Paulista Júlio de Mesquita Filho Câmpus de Botucatu Faculdade de Medicina, Botucatu, SP, Brazil
[4] Palliative Care Institute Liverpool, University of Liverpool, Liverpool, UK
[5] CEDAR—Center for Educational Development and Research in Health Sciences, University Medical Centre Groningen, Groningen, Netherlands
[6] Life and Health Sciences Research Institute, University of Minho, Braga, Portugal

**Acknowledgements** Palliative Care Institute Liverpool, the University of Liverpool for hosting and supporting the manuscript writing; Medical students and supervisors involved in data collection.

**Contributors** GG, SM and MAdC-F conceived and designed the study. GG and GAMdB collected the data. GG, DC-F and MAdC-F analysed the data. GG and MAdC-F were the major contributors in manuscript writing. DC-F, GAMdB and SM provided meaningful inputs and critical review of the manuscript. All authors read and approved the final manuscript.

**Funding** This work was supported by CAPES, grant number 88881.188776/2018-01 (PhD sandwich scholarship—Edictal 47/2017) to the first author for analysis and manuscript writing.

**Competing interests** GAMdB is a supervisor in MS2 and participates in collecting data.

**Patient consent for publication** Not required.

**Ethics approval** This research was conducted following the Declaration of Helsinki. We assured any student who was uncomfortable with the research enquiring would not feel obliged to participate. Themes related to death could be emotive and/or sensitive; thus, students could contact the researchers to receive support or referral if/as required. The Research Ethics Committee of both medical schools approved the study (58198016.4.0000.5404/2016; 58198016.4.3001.5411/2016).

**Provenance and peer review** Not commissioned; externally peer reviewed.

**Data availability statement**  Data are available upon reasonable request. The data from scales and questionnaires are available under request to Guilherme Gryschek (ORCID: https://orcid.org/0000-0002-1228-6371) for reasonable research use.

**ORCID iDs**
Dario Cecilio-Fernandes http://orcid.org/0000-0002-8746-1680
Marco Antonio de Carvalho-Filho http://orcid.org/0000-0001-7008-4092

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
