## [Reviewer comments · BMJ Open]

ARTICLE DETAILS

TITLE (PROVISIONAL)	Examining the effect of non-specialized clinical rotations upon medical students' Thanatophobia and Self-efficacy in Palliative Care: a prospective observational study in two medical schools
AUTHORS	Gryschek, Guilherme; Cecilio-Fernandes, Dario; Barros, Guilherme; Mason, Stephen; de Carvalho-Filho, Marco Antonio

VERSION 1 – REVIEW

REVIEWER	Junji Haruta Keio University in Japan
REVIEW RETURNED	21-Jun-2020

GENERAL COMMENTS	This research is a very important topic because of the growing needs for palliative care education. The comparison of the effects of palliative care education on self-efficacy and thanatophobia is interesting between two difficult medical universities. However, the logic of the study is unclear. As an educational study, there are several issues that need to be addressed and I think significant changes are needed. Here are some of the recommended improvements Major The most difficult part of this study is that the students of MS1 (Medical School1) which has a self -feedback or reflexive learning to improve their efficacy in the rotation, seems to increase the score of the self-efficacy compared to those who of MS2 (Medical School2) including a lesser such educational methods. In other words, regardless of palliative care, self-efficacy-focused educational methods appears to enhance the scores of the self-efficacy simply using “The Self-efficacy in Palliative Care”. I wonder what are you trying to show as what's new. In addition, the findings of the study should be compared to previous findings considering the mechanism and logical explanation in the discussion section. Here are some specific comments. Title. 1. As title, the authors wrote “Examining the effect of specialized and non-specialized clinical rotations...”. I'm not sure if this is robust comparison between rotations. Strictly speaking, the both education are not specific to palliative care. It is recommended that the title should be changed so that readers will not be misled. Introduction. 2. Social cognitive theory, which is the key to this study, is used as the basis for learning about palliative care. Explicitly stating the reasons for using it as a theoretical perspective would help the reader understand it logically.
---

	3. After presenting a review of previous reviews of education methods and assessments for medical students in palliative care, the authors should clarify the positioning of this study. 4. P5 L2-4 That effective learning is mostly related to how any given scenario is modulated by students and supervisors to optimize the learning process through different lenses. The authors should describe the reasons why the authors thought it based on the literature evidence. Methods Design, setting, participants 5. How many medical students of each rotate were in MS1 and MS2 in 2017? How long the difference in learning experience between the first and last group of the rotations in MS1 and MS2? Following topics should be described because I thought they might be the confounding factors of this assessments. 6. p6 Medical School1 What types of palliative experience do supervisors have? How long do supervisors learn/engage in medical education? Please describe how proficient supervisors are in medical education. In addition, in Table 1, the authors described “case discussions of the complex and critical patients” as teaching highlights. Please clarify how students participate in such educational sessions. It would be easier to understand the curriculum of the rotation as a whole if the authors showed the examples of the entire 8-week in Appendix and elsewhere. 7. p7 Medical School2 Similarly, how long do supervisors learn/engage in medical education? Please describe how proficient supervisors are in medical education. In addition, in Table 1, the authors described “case discussions of chronic patients” as teaching highlights. Please clarify how students participate in such educational sessions. The authors should clarify the extent to which students contribute to the multidisciplinary team. If medical students have opportunities to accept the feedbacks for healthcare professionals in the work place, please clarify them. Similarly, it would be easier to understand the curriculum of the rotation as a whole if the authors showed the examples of the entire 3-week in Appendix and elsewhere. 8. Table 1 shows that the Medical School 2 rotations were described as 3.5weeks, but the body says 3 weeks. Thus, please make it consistent. Procedure. 7. If the authors have explicitly stated that the possibility of affecting medical students’ response does not affect their grade, please describe it. 8. If the main author (GG) explained to students the research goals in the study on the first day of MS1 and MS2, then for medical students in one university, another university faculty member has explained the research. In that case, a relationship has not been made enough trust to collect research data between the faculty and the student. Please describe any psychological considerations. Data analysis 9. P10 L5
--	---

	The authors stated that Cohen'd and effect size were checked, but Cohen 'd is understood to be the same thing as effect size. If Cohen'd and effect size are different scores, please specify about what effect size is. 10. Please describe the way of the statistical analysis in making multiple comparisons. Results 11. We cannot find the results of VAS listed in the method. 12. p11 L3 The authors stated that “not in MS2 (p<0.05)”, but I guess that the authors might change the significance probability for multiple comparisons. Please describe it. Discussion I think the discussion section need to be significantly revised. 13.The first six lines was described the context of topics related to this research. In general, the findings of this study should be described in the first line of the discussion section. I recommend that the authors should do. 14. The authors should discuss the findings of this research based on the findings of previous papers. It is recommended that the mechanism and logical explanation about the findings should be described clearly. 15.P14L3 The authors stated “Although MS1 did not have specific activities devoted to palliative care”, but Table 1 showed the breaking bad news and end of life ethics discussion in MS1. In this context, what is specific activities devoted to palliative care? Please clarify it. Likewise, please specify with evidence what the authors mean by referring to palliative care in MS2 including specific activities devoted to palliative care. 16.p14 The authors stated that supervisor and reflection are effective for medical students in MS1, but the evidence for this is lack. The readers cannot understand the reasons. Especially since the comparison is based on different curricula, teachers, and learning environments to date, this conclusion is a bit premature to give it to you. Please carefully discuss what can be said about the findings of this study. Strengths and limitations 17. p15 L8 The authors stated “both groups had improvements in their self-efficacy, even when did not have access to specialized training in PC”, but so far MS2 has been described as specific activities devoted to palliative care as above. I'm confused. Please provide a consistent explanation. 18. I thought the results of this study were associated with the length of the rotation, the impact of other rotations, the timing of the study etc. As limitations, the authors should describe the confounding factors. Conclusion 19. Please revise your comment above. Minor P5 L4 20. “the learning process” is duplicated in P5L4.
--	--

REVIEWER	Dr Barnaby Hole University of Bristol, UK
REVIEW RETURNED	22-Jun-2020

GENERAL COMMENTS	Dear team, Thank you for allowing me to read this work. It is great to see analysis of how we can better integrate palliative care into routine undergraduate education. I think this paper should be published, but I have a few recommendations for improvement: Firstly, and most importantly, I think much more information is needed to understand what the students actually got at the two medical schools, such that a reader can begin to imagine what their experiences were and how these might have driven changes in their scores. I read the paper as advocating for non-specialist PC training embedded in general clinical education - if so, the authors should endeavour to describe the training provided at MS1 sufficiently well, such that another medical school could begin to develop their curriculum around what is delivered, or another experimenter could design a study to test whether this works. How many instructors were involved? What specialities are they trained in? Are these people authors on this paper (and if so, this should be described in the text and noted as a conflict of interest). Is a structured approach used to guide the integration of palliative care principles into case discussions, or is this a completely flexible process? How many dying patients are students in the MS1 rotation meeting and what are they dying from? Do they see their deaths? etc. etc. Similarly for MS2. Without enough information the paper becomes an exercise in comparing the results of one black box with another. I think this deserves substantial description and the authors could substantially reduce the word count of the introduction and conclusion to make space. Alternatively, curricula, learning outcomes and descriptions of student experiences could be mounted as appendices. Secondly, and linked with the above, greater care should be made by the authors to reduce the implication that causal inferences can be drawn about the effectiveness of the teaching and the different effectiveness between the schools. It is quite likely that a substantial aspect of the changes recorded was driven by passage of time, clinical exposure and preparation for their experience of post-graduate practice. In the absence of a "control", one cannot relate these changes to the palliative care aspect of their training (which as described in the table, makes up a small amount of their total education in both settings). Furthermore, the two schools likely recruit very different populations of students, clinicians, trainers etc. and have different approaches to education, professionalism, acquisition of skills/attitudes/knowledge etc. such that the differences at baseline and in experience for the two groups may be substantial, irrespective of the approach taken in palliative care education. This second suggestion requires alteration mainly to the discussion and conclusion, but should not be cursory. It is very tempting to draw causal inference, but to do so would be unfounded. It is especially inappropriate to comment on comparative effectiveness of the two approaches. In a nutshell, the work shows that SEPC and TS scores improve (more than would be expected as random variation between two random samples) for 6th year students at two medical schools during their
---

	senior clinical attachments (as described). Whilst differences at baseline and in the magnitude of change are seen between the schools, the study is not designed (or able) to ascertain whether any of the changes seen relate to the underlying approach taken in palliative care education. Interventional research would be needed to address this question. I will gladly review a revised version of the manuscript. Best, Barny
--	--

VERSION 1 – AUTHOR RESPONSE

Reviewer(s)' Comments to Author:

Reviewer: 1

Reviewer Name: Junji Haruta

Institution and Country: Keio University in Japan

This research is a very important topic because of the growing needs for palliative care education. The comparison of the effects of palliative care education on self-efficacy and thanatophobia is interesting between two difficult medical universities. However, the logic of the study is unclear. As an educational study, there are several issues that need to be addressed and I think significant changes are needed. Here are some of the recommended improvements

Comments: we appreciate your thoughtful reading of our manuscript. Palliative care education is still in its early stages in Brazil, and we look forward to contributing to the development of this field. Thank you for your recommendations that surely will improve our manuscript.

Major

The most difficult part of this study is that the students of MS1 (Medical School1) which has a self-feedback or reflexive learning to improve their efficacy in the rotation, seems to increase the score of the self-efficacy compared to those who of MS2 (Medical School2) including a lesser such educational methods. In other words, regardless of palliative care, self-efficacy-focused educational methods appears to enhance the scores of the self-efficacy simply using "The Self-efficacy in Palliative Care". I wonder what are you trying to show as what's new.

Answer: We agree with the reviewer that providing opportunities for self-reflection could improve measures of self-efficacy in general. However, taking care of dying patients without professional and emotional support can lead to compassionate fatigue, emotional detachment, and even burnout. Our study shows that this support can be achieved in general rotations that not count on the participation of specialists or specialized centers. This is an important finding to encourage clinical teachers in countries with low resources to fight for including palliative care training in underserved contexts. Nevertheless, we included in the limitation session a commentary on the possibility of general improvement of self-efficacy.

Adjustment (Limitations session, p16): "Also, improvements in self-efficacy, in general, can be related to the opportunity given to medical students to reflect on their clinical experiences under the guidance of motivated facilitators (5). This general improvement could have also been extrapolated to Palliative Care competencies (6)."

In addition, the findings of the study should be compared to previous findings considering the mechanism and logical explanation in the discussion section.

Answer: thank you for this comment. We included in Discussion section a comparison with a similar previous study.

Adjustment (Discussion section, p13): "In both schools, the placements brought improvements in students' self-efficacy in palliative care, in all subscales, as expected after a learning experience that includes PC topics and discussions. Another study with UK medical students had similar results (8). Nevertheless, in the UK study, the effect size magnitude was larger in almost every subscale of SEPC, although the pre-test results were similar. One explanation for this could be the specialized PC placement the UK students had. There are other factors to be considered: time dedicated to PC training, specialized supervisors and services, placement length and the learning environment. In this case, a specialized placement, focused on training PC competencies, could bring development to students' self-efficacy and decrease their thanatophobia."

Here are some specific comments.

Title.

1. As title, the authors wrote "Examining the effect of specialized and non-specialized clinical rotations...". I'm not sure if this is robust comparison between rotations. Strictly speaking, the both education are not specific to palliative care. It is recommended that the title should be changed so that readers will not be misled.

Answer: we agree with the reviewer and corrected the title. Also, we changed the arguments in the main manuscript to adjust to this line of reasoning.

Adjustment (Title): "Examining the effect of non-specialized clinical rotations upon medical students' Thanatophobia and Self-efficacy in Palliative Care: a prospective observational study in two Brazilian medical schools."

Introduction.

2. Social cognitive theory, which is the key to this study, is used as the basis for learning about palliative care. Explicitly stating the reasons for using it as a theoretical perspective would help the reader understand it logically.

Answer: In this study, the social cognitive theory is used as a theoretical framework that explains how learning (knowledge, skills, and attitudes) is developed, assimilated, and consolidated. Hence, as a theory of how people learn, assessing the key constructs within social cognitive theory will enable an assessment of the effects of training programmes – and therefore provides a theoretical driver for the design of the assessment instruments used in this study.

Adjustment (Introduction, p3): we included an additional statement to reinforce the argument above: "Social cognitive theory has been employed in studies to assess educational interventions, particularly where practical assessment of the development of knowledge, skills, and attitudes are difficult to apply efficiently at scale (7,8)."

3. After presenting a review of previous reviews of education methods and assessments for medical students in palliative care, the authors should clarify the positioning of this study.

Answer: we inserted some additional information to clarify our study's medical schools' context related to palliative care education.

Adjustment (Introduction, p5, 6): "Although medical schools are recommended to include palliative care in their curricula, the teaching and assessment strategies are heterogenous, and many schools do not have access to PC specialized services for students' placements (12,13). Examples of

specialized services are hospices, wards, and outpatients' clinics run by palliative medicine specialists. While in Europe and other countries the palliative care have wider offer and students training are based in specialized services, in Latin America that is not the case (14,15). For example, in Brazil, there are less than 200 specialized PC services, and many are not linked to medical schools (16). In practical terms, most of medical students in Brazil do not have access to specialized training settings, which is the case of the medical schools in our research (17). In Brazilian medical school, the overloaded curricula and the lack of specialized palliative care teachers hinder palliative care undergraduate education (13)."

4. P5 L2-4

That effective learning is mostly related to how any given scenario is modulated by students and supervisors to optimize the learning process through different lenses. The authors should describe the reasons why the authors thought it based on the literature evidence.

Answer: We appreciate your comment. We included references that support the theoretical explanation of the adult learning process.

Adjustment (Introduction, p5): "Engagement and learning within a specialized practice may be considered ideal for training purposes. However, effective learning is mostly related to how any given scenario is modulated by students and supervisors to optimize the learning process through different lenses (18,19). Among these lenses, the Kolb's cycle offers a compelling insight into understanding the learning process triggered by practical experiences (19)."

Methods

Design, setting, participants

5. How many medical students of each rotate were in MS1 and MS2 in 2017? How long the difference in learning experience between the first and last group of the rotations in MS1 and MS2?

Answer: we inserted and corrected this information in the Design, Setting, and Participants subsection.

Adjustment (Methods section, Design, Setting and Participants subsection, p7, 8): "Medical School 1 (MS1): final year students undertake an 8-week mandatory emergency and critical care placement, with 14 students per group. The first group took part in the research during the beginning of the final year, in January 2017, and the last one in September 2017." and

"Medical School 2 (MS2): final year medical students undertake a 6-week mandatory rotation in Anaesthesiology and pain therapy, in groups of 12 students. The first group answered to this research in February 2017 and the last one in September 2017.

Following topics should be described because I thought they might be the confounding factors of this assessments."

6. p6 Medical School1

What types of palliative experience do supervisors have? How long do supervisors learn/engage in medical education? Please describe how proficient supervisors are in medical education. In addition, in Table 1, the authors described "case discussions of the complex and critical patients" as teaching highlights. Please clarify how students participate in such educational sessions. It would be easier to understand the curriculum of the rotation as a whole if the authors showed the examples of the entire 8-week in Appendix and elsewhere.

Answer: we decided to show the characteristics of both rotations in the appendix section and keep the main text as concise as possible.

Adjustment (Appendix):

“Medical Schools` Placements

In Medical School 1, students rotate in three different clinical environments. In the first 4 weeks, they stay in the Emergency Department. For an international audience, it is important to explain that Emergency Departments in Brazil are chaotic environments that are always working 4-5 times above their capacities. For instance, in medical school 1, the Emergency Department was planned to have 25 patients in observation, but end up admitting more than 100. Ideally, patients should stay there for 24 hours maximum, but occasionally they stay there for up to two weeks. Even patient who need intensive care can stay in the Emergency Department for up to 5 days waiting for a vacancy inside the hospital. Students engage actively in the process of care and take direct responsibility for patients, always under the supervision of an experienced physician. In the following 4 weeks, students stay for 2 weeks in an ICU, and 2 weeks in an internal medicine with a high level of complexity, both devoted to patients admitted through the Emergency Department. In both, the internal medicine ward and ICU, in the morning, students are responsible for presenting the case of the patients during the daily clinical round and for organizing and implementing the therapeutic plan that is discussed together with the multi-professional team. On Mondays and Fridays mornings, after the round, students engage in a clinical ethics discussion about one of the patients they are caring for. In the afternoon, on Wednesdays and Thursdays, students engage in simulation activities dedicated to address communication issues and develop emotional competence. Every Friday afternoon, students have discussions about death and dying and palliative care. The structure of the weeks in the mentioned rotations are shown below.

MS1 – Week Planning	Monday	Tuesday	Wednesday	Thursday	Friday
Weeks 1 to 4 Morning	Emergency Department	Emergency Department	Emergency Department	Emergency Department	Emergency Department
Afternoon	Emergency Department	Emergency Department	Emergency Department	Emergency Department	Emergency Department
Weeks 4 to 8 Morning	ICU Clinical Round Ethics Discussion	ICU Clinical Round	ICU Clinical Round	ICU Clinical Round Ethics Discussion	ICU Clinical Round
Afternoon	Lectures	Lectures	Simulation Sessions	Simulation Sessions	Discussions about death and dying

	Medical School 1 – Placement Summary
Placement	Emergency and Critical Care Medicine
Length	8 weeks
Program hours	440 hours

Learning Outcomes related to Palliative Care	 - Identify terminal patients and understand the moment to shift from a predominant curative approach to a predominant palliative approach. - Provide end-of-life care to terminal patients - Understand the concepts of orotodhanasia and dysthanasia - Understand and apply the ethical principles of surrogate decision-making - Understand and apply the principles of interprofessional collaboration in the context of end-of-life care - Understand how the different cultural perspectives related to death and dying impact on end-of-life care - Understand how our own perspectives on death and dying may interfere with the end-of-life care that we provide to our patients - Develop patient-centered communication based on presence, improvisation, and empathy.
Groups size	14 students
Learning activities	 - Daily rounds to: emergency department and critical care wards; - Case discussion sessions; - High fidelity simulation sessions: emergency room and consultation scenarios; - Point-of-care ultrasound classes and bedside training; - Remote learning: digital educational plataform Moodle; - Mechanical ventilation classes; - Palliative care classes; - Palliative care ethics discussions;
Palliative care topics	 - Breaking bad news; - End-of-life ethics; - Thanatology;
Total Supervisors	All Medical doctors  - Emergency department: 8 - Internal Medicine Ward: 2 - Intensive Care Unit: 2
Supervisors experience in Medical Education	The course coordinator is a researcher in medical education, and 2 of the eight supervisors have a Ph.D. in medical education focused on developing new active learning strategies. All the supervisors have consistently participated in medical education congresses, presenting both research and educational innovations.
Supervisors experience in Palliative Care	Lecturer in Palliative care topics End of life care ethics discussions in weekly basis;
Educational Methods	 - Daily wards rounds; - Blended methods: classes and remote learning; - Bedside teaching; - Simulation sessions; - Clinical case discussions;

Learning activities related to PC and end-of-life care	 - Clinical simulation targeting breaking bad news, communication skills, and emotional competence (4 sessions of 4h each); - End of Life Ethics discussions (4 sessions of 1 hour each);
Teaching highlights	 - Simulation and feedback - Reflexive learning - Case discussion of complex and critical patients
Multidisciplinary team coworking	Coworking close to other medical specialties: psychiatry, neurology, orthopedics and traumatology
Supervisors Feedback routine	Daily feedback in wards rounds and other learning activities, such as Moodle and simulation sessions. Grades mailing in the end of the placement with all evaluation results
Evaluation	Multidimensional evaluation  - Two theoretical classic tests: multiple choice and open questions - OSCE: clinical scenarios based in placement daily situations - Supervisors, peers and self-evaluation on attitudes and behavior

7. p7 Medical School2

Similarly, how long do supervisors learn/engage in medical education? Please describe how proficient supervisors are in medical education. In addition, in Table 1, the authors described "case discussions of chronic patients" as teaching highlights. Please clarify how students participate in such educational sessions. The authors should clarify the extent to which students contribute to the multidisciplinary team. If medical students have opportunities to accept the feedbacks for healthcare professionals in the work place, please clarify them. Similarly, it would be easier to understand the curriculum of the rotation as a whole if the authors showed the examples of the entire 3-week in Appendix and elsewhere.

Answer: we decided to show the characteristics of both rotations in the appendix section and to keep the main text as concise as possible.

Adjustment (Appendix):

"In Medical School 2 the Anesthesiology and Pain Medicine placement have a total of 280 hours of activities. About 140 hours of this workload is related to palliative care topics along pain medicine learning activities. The group of students are divided into sub-groups composed of 2 students to palliative care activities, which are performed during home visits, in wards and in outpatient clinics, assisting patients under palliative care. All these clinical-practical activities take place in multiprofessional settings with the presence of nurses, social workers and other professionals. At the same time there are theoretical activities in which subjects such as the treatment of acute and chronic pain, bases of palliative care and health communication are addressed, with an emphasis on bad news.

MS2 – Week Planning	Monday	Tuesday	Wednesday	Thursday	Friday
--------	---------	-----------	----------	--------

Weeks 1 to 3 Morning	Anesthesiology department	Anesthesiology department	Anesthesiology department	Anesthesiology department	Anesthesiology department
Afternoon	Anesthesiology department	Anesthesiology department	Anesthesiology department	Anesthesiology department	Anesthesiology department
Weeks 3 to 6 Morning	Pain and Palliative Care Outpatient Clinic	Pain and Palliative Care Outpatient Clinic	Pain and Palliative Care Outpatient Clinic	Pain and Palliative Care Outpatient Clinic	Pain and Palliative Care Outpatient Clinic
Afternoon	Lecture	Lecture	Clinical case discussion	Pain and Palliative Care Outpatient Clinic	Homecare

	Medical School 2 – Placement Summary
Clinical placement	Anesthesiology and Pain Medicine
Length	6 weeks
Total program hours	280 hours
Learning Outcomes related to Palliative Care	 - Identify patients who can benefit from palliative care - Understand the concepts of palliative care - Understand the importance of adequate communication in health and symptoms management - Evaluate and manage acute and chronic pain
Groups size	12 students
Learning activities	 - Traditional and interactive lectures using Kahoot (gamebased learning platform) - Patient care in outpatient clinic, in the hospital yards and home care - Clinical cases discussion
Palliative care topics	 - Palliative care concepts - Communication in palliative care - Pain and symptom management
Total Supervisors	 - Medical doctors: 2 - Registered Nurses: 2
Supervisors experience in Medical Education	Medical Doctors are PhD, working as clinical teachers at the medical school for more than 10 years

Supervisors experience in Palliative Care	Medical Doctors are board-certificated palliative care specialists Registered Nurses have no specific training
Educational Methods	 - Lectures - Clinical case discussions - Training procedures skills - Clinical exposure to palliative patient care
Learning activities related to PC and end-of-life care	 - PC lectures (8h): concepts, communications skills, symptoms management - Clinical discussions of outpatient patients (20h) - Homecare visits (4h)
Teaching highlights	 - Lectures - Case discussions of chronic patients - Working in a multi-professional healthcare team specialized in Palliative Care
Multidisciplinary team coworking	Discussion of clinical home care cases with the participation of registered nurse, social worker, psychologist and nutritionist. Physical therapists are available under referral
Supervisors Feedback routine	During daily practical activities
Evaluation	Classical style tests and clinical skill practical evaluation using simulated setting activities

8. Table 1 shows that the Medical School 2 rotations were described as 3.5weeks, but the body says 3 weeks. Thus, please make it consistent.

Answer: We thank the reviewer for noticing this inconsistency. We corrected it to 6 weeks in body and table, which is the total time of the placement, discriminating how activities related to PC are distributed.. We rechecked and corrected the information.

Adjustment (Methods section, Design, Setting and Participants subsection, p8): “Medical School 2 (MS2): final year medical students undertake a 6-week mandatory rotation in Anaesthesiology and pain therapy, in groups of 12 students. The first group answered to this research in February 2017 and the last one in September 2017. Significant part of the training in this rotation is dedicated to Anaesthesia abilities training. Within this period, students also have 32 hours dedicated to specific PC topics and experiences, spread in 3 weeks.”

Procedure.

9. If the authors have explicitly stated that the possibility of affecting medical students' response does not affect their grade, please describe it.

Answer: we inserted this description in Procedure Subsection because it was part of the instructions given to students previously to data collection.

Adjustment (Methods section, Procedure subsection, p9): “Two authors, GG and GAMB

(supervisor in MS2), personally invited the students during the introductory activities of each rotation in MS1 and MS2, respectively. Both explained to students the research goals, the nature of the instruments that would be applied and guaranteed the anonymity of the participants. Authors also assured that participating in the research would not affect students' grades in that rotation, trying to minimize students' perception of being evaluated and its interference in data provided."

10. If the main author (GG) explained to students the research goals in the study on the first day of MS1 and MS2, then for medical students in one university, another university faculty member has explained the research. In that case, a relationship has not been made enough trust to collect research data between the faculty and the student. Please describe any psychological considerations.

Answer: In MS2, another author, GAMB, collected the data. GAMB is a clinical teacher at MS2. We rephrased the methods section and included this limitation in our manuscript.

Adjustment (Methods section, Procedure subsection, p9): "Two authors, GG and GAMB (supervisor in MS2), personally invited the students during the introductory activities of each rotation in MS1 and MS2, respectively. Both explained to students the research goals, the nature of the instruments that would be applied and guaranteed the anonymity of the participants."

(Strengths and limitations section, p16): "Additionally, in MS2, as one of the researchers was supervisor, a socially desirable answer from the participants cannot be excluded."

Data analysis

11. P10 L5

The authors stated that Cohen's d and effect size were checked, but Cohen's d is understood to be the same thing as effect size. If Cohen's d and effect size are different scores, please specify about what effect size is.

Answer: We have rephrased this sentence.

Adjustment (Methods section, Data analysis subsection, p10): "Effect size was calculated by Cohen's d and used for checking the amplitude of changes in pre- and post-test scores, providing an index to assess the magnitude of the difference detected on t-test results."

12.

Please describe the way of the statistical analysis in making multiple comparisons.

Answer: Thank you for your comment. Indeed, we did not correct the alpha value for multiple comparisons. We have now corrected the alpha value using Bonferroni.

Adjustment (Methods section, Data analysis subsection, p10): "Bonferroni was used to correct the alpha value. Therefore, we used alpha value of 0.125 for all analyses."

Results

13. We cannot find the results of VAS listed in the method.

Answer: we did not understand this; is it possible to clarify to what 'VAS' refers to, as we did not list that abbreviation in the manuscript?

14. p11 L3

The authors stated that "not in MS2 ($p < 0.05$)", but I guess that the authors might change the significance probability for multiple comparisons. Please describe it.

Answer: All the significance values were correct using Bonferroni.

Adjustment (Methods section, Data analysis subsection, p10): "Bonferroni was used to correct the alpha value. Therefore, we used alpha value of 0.125 for all analyses."

Discussion

I think the discussion section need to be significantly revised.

15. The first six lines was described the context of topics related to this research. In general, the findings of this study should be described in the first line of the discussion section. I recommend that the authors should do.

Answer: we thank you for this comment! We rewrite this section in several points to make better sense related to our results and hypothesis. We hope now it reads clear and consistent. As suggested, we started with a summary of our findings.

Adjustment (Discussion section, p12, 13): "In this study, the SEPC and TS measured significant improvements in students' self-efficacy in palliative care and thanatophobia levels, after the placements analysed, except in MS2, in which TS changes were not significant. These results support that non-specialized rotation can provide PC training with effects in students' self-efficacy in PC and thanatophobia. Most of the students in both schools informed some previous experience with dying people, or as a personal experience or during their professional training."

16. The authors should discuss the findings of this research based on the findings of previous papers. It is recommended that the mechanism and logical explanation about the findings should be described clearly.

Answer: Thank you this suggestion! We included a direct comparison to a similar study in a different context.

Adjustment (Discussion section, p13): "In both schools, the placements brought improvements in students' self-efficacy in palliative care, in all subscales, as expected after a learning experience that includes PC topics and discussions. Another study with UK medical students had similar results (8). Nevertheless, in the UK study, the effect size magnitude was larger in almost every subscale of SEPC, although the pre-test results were similar. One explanation for this could be the specialized PC placement the UK students had. There are other factors to be considered: time dedicated to PC training, specialized supervisors and services, placement length and the learning environment. In this case, a specialized placement, focused on training PC competencies, could bring development to students' self-efficacy and decrease their thanatophobia. In MS1, a decreasing in TS was observed, but not in MS2. Maybe this is a consequence of the experience in MS1 during the placement is related to more experience with critical and dying patients (see Appendix)."

17.P14L3

The authors stated "Although MS1 did not have specific activities devoted to palliative care", but Table 1 showed the breaking bad news and end of life ethics discussion in MS1. In this context, what is specific activities devoted to palliative care? Please clarify it. Likewise, please specify with evidence what the authors mean by referring to palliative care in MS2 including specific activities devoted to palliative care.

Answer: This comment threw us to a thoughtful reconsideration of our results, context and arguments. In our context, as PC practices are not common, even for us it was not clear what we should consider specialized or non-specialized PC training. Throughout the manuscript we rewrote several aspects based in this new understand, trying to clarify this misunderstanding. So, both placements are non-specialized PC rotations, but both offer activities related to PC. Please, read the new version of this manuscript keeping this major adjustment in mind.

Adjustment (Introduction section, p5): "Thus, we hypothesize that non-specialized centers can provide a proper learning environment to offer meaningful and productive clinical experiences, which could be capable of grounding the improvement of self-efficacy in PC and the decreasing of thanatophobia levels (19)."

(Discussion section, p14): "Our study showed that non-specialized rotations, which include specific topics in PC, were capable of improving students` self-efficacy in PC and decreasing their thanatophobia levels."

18.p14

The authors stated that supervisor and reflection are effective for medical students in MS1, but the evidence for this is lack. The readers cannot understand the reasons. Especially since the comparison is based on different curricula, teachers, and learning environments to date, this conclusion is a bit premature to give it to you. Please carefully discuss what can be said about the findings of this study.

Answer: we considered your point and decided to excluded all the results related to compare medical schools as we really lack of evidences to make those statements. Instead, we propose explanations that apply to both schools results.

Adjustment (Discussion section, p13): "In both schools, the placements brought improvements in students` self-efficacy in palliative care, in all subscales, as expected after a learning experience that includes PC topics and discussions. Another study with UK medical students had similar results (8). Nevertheless, in the UK study, the effect size magnitude was larger in almost every subscale of SEPC, although the pre-test results were similar. One explanation for this could be the specialized PC placement the UK students had. There are other factors to be considered: time dedicated to PC training, specialized supervisors and services, placement length and the learning environment. In this case, a specialized placement, focused on training PC competencies, could bring development to students` self-efficacy and decrease their thanatophobia. In MS1, a decreasing in TS was observed, but not in MS2. Maybe this is a consequence of the experience in MS1 during the placement is related to more experience with critical and dying patients (see Appendix)."

Strengths and limitations

19. p15 L8

The authors stated "both groups had improvements in their self-efficacy, even when did not have access to specialized training in PC", but so far MS2 has been described as specific activities devoted to palliative care as above. I'm confused. Please provide a consistent explanation.

Answer: we are sorry about this confusing statement, it was excluded and we rewrote the limitations sections to reflect the adjustments we made throughout the manuscript.

Adjustment (Strengths and limitations section, p15): "This study had a consistent methodological approach and involved two different medical schools, with different undergraduate curricula, which included PC training in non-specialized rotations. We demonstrated that, although the two groups of

students had different approaches in the sense of learning PC, both groups had improvements in their self-efficacy, even when students did not have access to specialized PC services.”

20. I thought the results of this study were associated with the length of the rotation, the impact of other rotations, the timing of the study etc. As limitations, the authors should describe the confounding factors.

Answer: we adjust this section including these well pointed limitations.

Adjustment (Strengths and limitations section, p15): “Nevertheless, we should acknowledge some limitations in our study. Firstly, we did not assess students’ conceptual knowledge of PC, neither the effects of other factors as placement length or learning approaches. In this sense, as we did not control other factors, we cannot conclude if the changes in scales are related to daily placements’ clinical situations or to specific teaching approaches. An interventional study with control groups would be more appropriate to verify if and how the different educational approaches could have different impacts on self-efficacy in palliative care and thanatophobia levels.”

Conclusion

11. Please revise your comment above.

Answer: we consider that, now, the conclusion is aligned with the discussion.

Adjustment (Conclusion section, p16):

“Implications for practice and next steps

In a context of an overloaded curriculum and a lack of departments and services of palliative medicine in medical schools, our findings suggest that inserting PC topics in different nonspecialized rotations could benefit the development of medical students self-efficacy in PC and reduce their thanatophobia. Thus, a vertical insertion of PC concepts and practices along the course, in different disciplines and rotations, could address PC learning needs for undergraduate students, even without specialized PC services for medical training.

We still need to understand the impact of implementing PC training on clinical rotations on students’ declarative and procedural knowledge acquisition. It is also essential to clarify what are the teaching methodologies related to better development of PC competencies in undergraduate medical students.

Conclusion

Our findings support that non-specialized rotations can include PC basic competencies as part of the training. Medical schools could prepare future doctors to provide PC in any area it is required by embracing palliative care as a core competence, indispensable for becoming a doctor.”

Minor

P5 L4

20. "the learning process" is duplicated in P5L4.

Answer: thank you, we corrected in main manuscript.

Reviewer: 2

Reviewer Name: Dr Barnaby Hole

Institution and Country: University of Bristol, UK

Dear team,

Thank you for allowing me to read this work. It is great to see analysis of how we can better integrate palliative care into routine undergraduate education. I think this paper should be published, but I have a few recommendations for improvement:

Comments: Thank you for the opportunity to review and update our manuscript! We appreciate your considerations and made efforts to review and change the aspects you pointed out.

Firstly, and most importantly, I think much more information is needed to understand what the students actually got at the two medical schools, such that a reader can begin to imagine what their experiences were and how these might have driven changes in their scores.

Answer: we considered important to include this information. We decided to give more details in appendix.

Adjustment (Appendix):

“Medical Schools` Placements

In Medical School 1, students rotate in three different clinical environments. In the first 4 weeks, they stay in the Emergency Department. For an international audience, it is important to explain that Emergency Departments in Brazil are chaotic environments that are always working 4-5 times above their capacities. For instance, in medical school 1, the Emergency Department was planned to have 25 patients in observation, but end up admitting more than 100. Ideally, patients should stay there for 24 hours maximum, but occasionally they stay there for up to two weeks. Even patient who need intensive care can stay in the Emergency Department for up to 5 days waiting for a vacancy inside the hospital. Students engage actively in the process of care and take direct responsibility for patients, always under the supervision of an experienced physician. In the following 4 weeks, students stay for 2 weeks in an ICU, and 2 weeks in an internal medicine with a high level of complexity, both devoted to patients admitted through the Emergency Department. In both, the internal medicine ward and ICU, in the morning, students are responsible for presenting the case of the patients during the daily clinical round and for organizing and implementing the therapeutic plan that is discussed together with the multi-professional team. On Mondays and Fridays mornings, after the round, students engage in a clinical ethics discussion about one of the patients they are caring for. In the afternoon, on Wednesdays and Thursdays, students engage in simulation activities dedicated to address communication issues and develop emotional competence. Every Friday afternoon, students have discussions about death and dying and palliative care. The structure of the weeks in the mentioned rotations are shown below.

MS1 – Week Planning	Monday	Tuesday	Wednesday	Thursday	Friday
Weeks 1 to 4 Morning	Emergency Department	Emergency Department	Emergency Department	Emergency Department	Emergency Department
Afternoon	Emergency Department	Emergency Department	Emergency Department	Emergency Department	Emergency Department
Weeks 4 to 8 Morning	ICU Clinical Round Ethics Discussion	ICU Clinical Round	ICU Clinical Round	ICU Clinical Round Ethics Discussion	ICU Clinical Round
Afternoon	Lectures	Lectures	Simulation Sessions	Simulation Sessions	Discussions about death and dying

	Medical School 1 – Placement Summary
Placement	Emergency and Critical Care Medicine
Length	8 weeks
Program hours	440 hours
Learning Outcomes related to Palliative Care	 - Identify terminal patients and understand the moment to shift from a predominant curative approach to a predominant palliative approach. - Provide end-of-life care to terminal patients - Understand the concepts of orotohanasia and dysthanasia - Understand and apply the ethical principles of surrogate decision-making - Understand and apply the principles of interprofessional collaboration in the context of end-of-life care - Understand how the different cultural perspectives related to death and dying impact on end-of-life care - Understand how our own perspectives on death and dying may interfere with the end-of-life care that we provide to our patients - Develop patient-centered communication based on presence, improvisation, and empathy.
Groups size	14 students
Learning activities	 - Daily rounds to: emergency department and critical care wards; - Case discussion sessions; - High fidelity simulation sessions: emergency room and consultation scenarios; - Point-of-care ultrasound classes and bedside training; - Remote learning: digital educational plataform Moodle; - Mechanical ventilation classes; - Palliative care classes; - Palliative care ethics discussions;
Palliative care topics	 - Breaking bad news; - End-of-life ethics; - Thanatology;
Total Supervisors	All Medical doctors  - Emergency department: 8 - Internal Medicine Ward: 2 - Intensive Care Unit: 2
Supervisors experience in Medical Education	The course coordinator is a researcher in medical education, and 2 of the eight supervisors have a Ph.D. in medical education focused on developing new active learning strategies. All the supervisors have consistently participated in medical education congresses, presenting both research and educational innovations.
Supervisors experience in Palliative Care	Lecturer in Palliative care topics End of life care ethics discussions in weekly basis;

Educational Methods	 - Daily wards rounds; - Blended methods: classes and remote learning; - Bedside teaching; - Simulation sessions; - Clinical case discussions;
Learning activities related to PC and end-of-life care	 - Clinical simulation targeting breaking bad news, communication skills, and emotional competence (4 sessions of 4h each); - End of Life Ethics discussions (4 sessions of 1 hour each);
Teaching highlights	 - Simulation and feedback - Reflexive learning - Case discussion of complex and critical patients
Multidisciplinary team coworking	Coworking close to other medical specialties: psychiatry, neurology, orthopedics and traumatology
Supervisors Feedback routine	Daily feedback in wards rounds and other learning activities, such as Moodle and simulation sessions. Grades mailing in the end of the placement with all evaluation results
Evaluation	Multidimensional evaluation  - Two theoretical classic tests: multiple choice and open questions - OSCE: clinical scenarios based in placement daily situations - Supervisors, peers and self-evaluation on attitudes and behavior

"In Medical School 2 the Anesthesiology and Pain Medicine placement have a total of 280 hours of activities. About 140 hours of this workload is related to palliative care topics along pain medicine learning activities. The group of students are divided into sub-groups composed of 2 students to palliative care activities, which are performed during home visits, in wards and in outpatient clinics, assisting patients under palliative care. All these clinical-practical activities take place in multiprofessional settings with the presence of nurses, social workers and other professionals. At the same time there are theoretical activities in which subjects such as the treatment of acute and chronic pain, bases of palliative care and health communication are addressed, with an emphasis on bad news.

MS2 – Week Planning	Monday	Tuesday	Wednesday	Thursday	Friday
Weeks 1 to 3	Anesthesiology department	Anesthesiology department	Anesthesiology department	Anesthesiology department	Anesthesiology department
Morning					
Afternoon	Anesthesiology department	Anesthesiology department	Anesthesiology department	Anesthesiology department	Anesthesiology department

Weeks 3 to 6 Morning	Pain and Palliative Care Outpatient Clinic	Pain and Palliative Care Outpatient Clinic	Pain and Palliative Care Outpatient Clinic	Pain and Palliative Care Outpatient Clinic	Pain and Palliative Care Outpatient Clinic
Afternoon	Lecture	Lecture	Clinical case discussion	Pain and Palliative Care Outpatient Clinic	Homecare

	Medical School 2 – Placement Summary
Clinical placement	Anesthesiology and Pain Medicine
Length	6 weeks
Total program hours	280 hours
Learning Outcomes related to Palliative Care	 - Identify patients who can benefit from palliative care - Understand the concepts of palliative care - Understand the importance of adequate communication in health and symptoms management - Evaluate and manage acute and chronic pain
Groups size	12 students
Learning activities	 - Traditional and interactive lectures using Kahoot (gamebased learning platform) - Patient care in outpatient clinic, in the hospital yards and home care - Clinical cases discussion
Palliative care topics	 - Palliative care concepts - Communication in palliative care - Pain and symptom management
Total Supervisors	 - Medical doctors: 2 - Registered Nurses: 2
Supervisors experience in Medical Education	Medical Doctors are PhD, working as clinical teachers at the medical school for more than 10 years
Supervisors experience in Palliative Care	Medical Doctors are board-certificated palliative care specialists Registered Nurses have no specific training
Educational Methods	 - Lectures - Clinical case discussions - Training procedures skills - Clinical exposure to palliative patient care

Learning activities related to PC and end-of-life care	 - PC lectures (8h): concepts, communications skills, symptoms management - Clinical discussions of outpatient patients (20h) - Homecare visits (4h)
Teaching highlights	 - Lectures - Case discussions of chronic patients - Working in a multi-professional healthcare team specialized in Palliative Care
Multidisciplinary team cworking	Discussion of clinical home care cases with the participation of registered nurse, social worker, psychologist and nutritionist. Physical therapists are available under referral
Supervisors Feedback routine	During daily practical activities
Evaluation	Classical style tests and clinical skill practical evaluation using simulated setting activities

I read the paper as advocating for non-specialist PC training embedded in general clinical education - if so, the authors should endeavour to describe the training provided at MS1 sufficiently well, such that another medical school could begin to develop their curriculum around what is delivered, or another experimenter could design a study to test whether this works. How many instructors were involved? What specialities are they trained in? Are these people authors on this paper (and if so, this should be described in the text and noted as a conflict of interest).

Answer: we considered important to include this information. We decided to give more details in appendix (see above). We also inserted a statement as conflict of interest and an additional comment related to that in Limitation section.

Adjustments:

(Strengths and limitations section, p15): "Additionally, in MS2, as one of the researchers was supervisor, a socially desirable answer from the participants cannot be excluded."

(Competing interests section, p17): "GAMB is a supervisor in MS2 and participate in collecting data."

(Methods section, Procedures subsection, p8): "Two authors, GG and GAMB (supervisor in MS2), personally invited the students during the introductory activities of each rotation in MS1 and MS2, respectively. Both explained to students the research goals, the nature of the instruments that would be applied and guaranteed the anonymity of the participants. Authors also assured that participating in the research would not affect students' grades in that rotation, trying to minimize students' perception of being evaluated and its interference in data provided."

Is a structured approach used to guide the integration of palliative care principles into case discussions, or is this a completely flexible process?

Answer: Cases discussions in wards, outpatient clinics and homecare are flexible process, based in that specific case and the interaction between students and supervisors, including the feedback. We described it better in the appendix.

How many dying patients are students in the MS1 rotation meeting and what are they dying from? Do they see their deaths? etc. etc.

Answer: it is hard to quantify exactly how many dying patients the students have contact during the rotation. We described the context of MS1 in appendix and context section.

Adjustment (Methods section, Design, Setting and Participants subsection, p7): "During this rotation, students engage in clinical activities for 11 hours daily, under the direct supervision of clinical teachers, and experience the care of chronically ill and dying patients. It is hard to quantify exactly how many dying patients the students have contact during the rotation. However, as the rotation take place in a non-surgical ICU, many patients have chronic diseases such as COPD, heart failure,

metastatic cancer etc. This ICU has 28 beds and a mortality rate of 30%. Unfortunately, in our context is still common that terminal patients (even those that should be considered for exclusive palliative care) receive their end-of-life care in the ICU or emergency room, as we lack palliative care services in the health network. Considering that, most of MS1 students are involved in caring for patients and assist them in the end of life and almost all testify the deaths of some of the patients during the rotation period. The clinical supervisors decided to include flexible PC discussions in bedside rounds due to the opportunities provided by this placement. Conversely, the supervisors provide formally structured discussions about end-of-life ethics in multi-professional meetings.”

Similarly for MS2. Without enough information the paper becomes an exercise in comparing the results of one black box with another. I think this deserves substantial description and the authors could substantially reduce the word count of the introduction and conclusion to make space. Alternatively, curricula, learning outcomes and descriptions of student experiences could be mounted as appendices.

Answer: We accepted your suggestion and included an appendix with detailed information about both rotations.

Secondly, and linked with the above, greater care should be made by the authors to reduce the implication that causal inferences can be drawn about the effectiveness of the teaching and the different effectiveness between the schools. It is quite likely that a substantial aspect of the changes recorded was driven by passage of time, clinical exposure and preparation for their experience of post-graduate practice. In the absence of a "control", one cannot relate these changes to the palliative care aspect of their training (which as described in the table, makes up a small amount of their total education in both settings). Furthermore, the two schools likely recruit very different populations of students, clinicians, trainers etc. and have different approaches to education, professionalism, acquisition of skills/attitudes/knowledge etc. such that the differences at baseline and in experience for the two groups may be substantial, irrespective of the approach taken in palliative care education. This second suggestion requires alteration mainly to the discussion and conclusion, but should not be cursory. It is very tempting to draw causal inference, but to do so would be unfounded. It is especially inappropriate to comment on comparative effectiveness of the two approaches. In a nutshell, the work shows that SEPC and TS scores improve (more than would be expected as random variation between two random samples) for 6th year students at two medical schools during their senior clinical attachments (as described). Whilst differences at baseline and in the magnitude of change are seen between the schools, the study is not designed (or able) to ascertain whether any of the changes seen relate the underlying approach taken in palliative care education. Interventional research would be needed to address this question.

Answer: we made a thoughtful consideration of your well-done argumentation and we agree with your comment. Recognizing a contextual mistake, we refocused the manuscript's main argument. Our study shows that this support can be achieved in general rotations that not count on the participations of specialists. This is an important finding to encourage clinical teachers in countries with low resources to fight for including palliative care trainings in underserved contexts. In order to clarify this point, we have made adjustments to show that in the Brazilian context, palliative care services are not common to specialized medical students training and, alternatively, palliative care could be inserted in different placements and successfully improve students` self-efficacy in palliative care. Also, we reformulated or excluded any comparison between schools, which is not in accord with our study's design to verify our hypothesis. In this sense, we rewrote extensively the discussion, limitations and conclusion sections. Hopefully, we addressed your comments.

Adjustments:

“Discussion

In this study, the SEPC and TS measured significant improvements in students' self-efficacy in palliative care and thanatophobia levels, after the placements analysed, except in MS2, in which TS

changes were not significant. Most of the students in both schools informed some previous experience with dying people, or as a personal experience or during their professional training. This could suggest that previous clinical placements give opportunities for caring of dying patients. Nevertheless, the placements analysed in this study gave to students an additional opportunity to care of dying patients and purposely learn and discuss PC topics.

In both schools, the placements brought improvements in students' self-efficacy in palliative care, in all subscales, as expected after a learning experience that includes PC topics and discussions. Another study with UK medical students had similar results (8). Nevertheless, in the UK study, the effect size magnitude was larger in almost every subscale of SEPC, although the pre-test results were similar. One explanation for this could be the specialized PC placement the UK students had. There are other factors to be considered: time dedicated to PC training, specialized supervisors and services, placement length and the learning environment. In this case, a specialized placement, focused on training PC competencies, could bring development to students' self-efficacy and decrease their thanatophobia. In MS1, a decreasing in TS was observed, but not in MS2. Maybe this is a consequence of the experience in MS1 during the placement is related to more experience with critical and dying patients (see Appendix).

Worldwide, medical educators have devised different strategies for PC training, and there is no 'gold standard' to guide best practices (12,28,29). Although training in specialized PC services could bring larger improvement in SEPC, these services are not common in Brazil and are not an option for training medical students. Including a PC specialized placement in the already busy and crowded medical curriculum poses a challenge for medical educators. The challenge has two edges; one is the scarcity of specialized services and professionals to facilitate training, particularly in the developing world (where the PC is still incipient and needs substantial development); the second is the lack of space in the curriculum.

Ultimately, clinicians in most specialties, such as internal medicine, geriatrics, oncology, emergency medicine, intensive care, primary care, paediatrics, and so on, will have PC patients among their caseloads. Although there can often be a lack of PC services, different disciplines and placements should include PC in their curricula and provide PC training for all medical students (30). Teachers could customize different clinical placements to modulate learning possibilities to students considering the resources available as showed in this study medical schools. By doing so, medical educators could ensure a psychological safety environment for learning, including feedback routines and the development of self-efficacy, which we considered as a fundamental aspect in schools we analysed (31). Our study showed that non-specialized rotations, which include specific topics in PC, were capable of improving students' self-efficacy in PC and decreasing their thanatophobia levels. Therefore, others aspects as learning environment and experience, feedback opportunities and the inclusion of PC as part of clinical caring, could play relevant elements for PC training in this context (19). We believe that this finding is related to opportunities to reflect on real end-of-life cases that were provided for the students during these placements. As anticipated by Kolb's experiential learning theory, the several occasions to reflect in and on the practice could have nurtured confidence and a readiness to practice PC (19). In fact, the social cognitive theory states that self-efficacy and outcome expectancy are both modulated by the feedback (perceived support) from supervisors, which will eventually, impact on the internalization of the desired behaviour (5). Possibly offering opportunities to experience and reflect, guided by qualified supervision with structured feedback, may be as important as having access to specialized PC centres for the learning of undergraduate students (5,6).

The challenge posed to medical educators is building up a meaningful learning process in complex workplaces in different clinical rotations, despite the lack of ideal circumstances (18). Clinical teachers need to nurture the learning experience of students embracing the contextual elements, even the negative ones, as teaching assets and not as pitfalls. Students will practice medicine in the real world and need to learn how to deal with the context of the workplace. Those with higher self-efficacy are more likely to reflect on their challenges and experiences to improve performance (5). In this sense, reflection is essential not only to learn but also to feel the urge to change and improve the healthcare

delivered to patients (8). Therefore, including structured PC discussions in clinical rotations may offer an opportunity for teachers not only to prepare students to provide proper palliative care but also to open the debate about the relevance of including PC into their future practice.

Strengths and limitations

This study had a consistent methodological approach and involved two different medical schools, with different undergraduate curricula, which included PC training in non-specialized rotations. We demonstrated that, although the two groups of students had different approaches in the sense of learning PC, both groups had improvements in their self-efficacy, even when students did not have access to specialized PC services.

Nevertheless, we should acknowledge some limitations in our study. Firstly, we did not assess students' conceptual knowledge of PC, neither the effects of other factors as placement length or learning approaches. In this sense, as we did not control other factors, we cannot conclude if the changes in scales are related to daily placements' clinical situations or to specific teaching approaches. An interventional study with control groups would be more appropriate to verify if and how the different educational approaches could have different impacts on self-efficacy in palliative care and thanatophobia levels.

Secondly, although the students have expressed more confidence in their performance, we cannot conclude that the improvements in SEPC and thanatophobia, observed in this study, will be persistent for the actual medical practice of the students. The proper assessment for this will require further evaluation over the new doctors in their practice.

Also, improvements in self-efficacy, in general, can be related to the opportunity given to medical students to reflect on their clinical experiences under the guidance of motivated facilitators (5). This general improvement could have also been extrapolated to Palliative Care competencies (6).

Additionally, in MS2, as one of the researchers was supervisor, a socially desirable answer from the participants cannot be excluded.

Furthermore, the findings in this study are limited to two Brazilian medical schools and any generalization should be cautious. Future studies are needed to verify whether our findings will be consistent across different contexts.

Implications for practice and next steps

In a context of an overloaded curriculum and a lack of departments and services of palliative medicine in medical schools, our findings suggest that inserting PC topics in different nonspecialized rotations could benefit the development of medical students self-efficacy in PC and reduce their thanatophobia. Thus, a vertical insertion of PC concepts and practices along the course, in different disciplines and rotations, could address PC learning needs for undergraduate students, even without specialized PC services for medical training.

We still need to understand the impact of implementing PC training on clinical rotations on students' declarative and procedural knowledge acquisition. It is also essential to clarify what are the teaching methodologies related to better development of PC competencies in undergraduate medical students.

Conclusion

Our findings support that non-specialized rotations can include PC basic competencies as part of the training. Medical schools could prepare future doctors to provide PC in any area it is required by embracing palliative care as a core competence, indispensable for becoming a doctor."

I will gladly review a revised version of the manuscript.

VERSION 2 – REVIEW

REVIEWER	Junji Haruta Keio University and Japan
REVIEW RETURNED	11-Aug-2020

GENERAL COMMENTS	This revised research is easy for the reader to understand and logically explained. Thus, this paper is worthy of publication, but there are a few minor revisions, so please check. Abstract The setting should be “Medical schools in Brazil”, not “Medical schools in South America since the method mentions Brazilian medical schools. Introduction P3. You should add a reference to the explanatory text of the outcome expectancy as following sentence. “Outcome expectancy refers to the perceived consequence of the target behaviour, together with the aggregate value this...” Data Analysis P10 The following statement should be listed as a reference for effect size. “The effect size was characterized as small, medium, large, or very large and evaluated the magnitude of change despite the sample size. “ The following statement should be listed as a reference for Bonferroni. If you took advantage of Bonferroni, could be P 0.0125 instead of 0.125? The reference in No. 28 as follows could not be found on the internet. “Data were analyzed using IBM-SPSS 21.0 (28).” 28. IBM Corp. IBM SPSS Statistics for Windows. Armonk, NY: IBM Corp.; 2012. Results P11-12 If the TS(Thanatophobia score) in MS2 difference listed in Table 2 is post-pre, then it seems to be a negative result. Also, when the text states that the TS in MS2 is not statistically decreased in MS2, you should state a specific p-value instead of stating $p < 0.05$.
---

REVIEWER	Dr Barnaby Hole University of Bristol
REVIEW RETURNED	24-Aug-2020

GENERAL COMMENTS	Dear team, Thank you for listening to and responding to completely to the reviewer comments. I wish you good fortune with this paper and your important educational work. Best, Barney
---

VERSION 2 – AUTHOR RESPONSE

Reviewer(s)' Comments to Author:

Reviewer: 1

Reviewer Name: Junji Haruta

Institution and Country: Keio University and Japan

Please state any competing interests or state 'None declared': None declared

Please leave your comments for the authors below

This revised research is easy for the reader to understand and logically explained. Thus, this paper is worthy of publication, but there are a few minor revisions, so please check.

Answer: we appreciate so much your contributions for our review, and we are grateful we could present a suitable manuscript.

1. Abstract

The setting should be "Medical schools in Brazil", not "Medical schools in South America since the method mentions Brazilian medical schools.

Answer: Thank you for the suggestion.

Adjustments (Abstract Section): Setting: Medical schools in Brazil.

2. Introduction

P3.

You should add a reference to the explanatory text of the outcome expectancy as following sentence. "Outcome expectancy refers to the perceived consequence of the target behavior, together with the aggregate value this..."

Answer: We inserted the references that refers more widely to outcome expectancy.

Adjustments: "Outcome expectancy" refers to the perceived consequence of the target behaviour, together with the aggregate value this behaviour brings to the student (5,6).

3. Data Analysis

P10

The following statement should be listed as a reference for effect size.

"The effect size was characterized as small, medium, large, or very large and evaluated the magnitude of change despite the sample size. "

Answer: We inserted an additional reference, as follows.

Adjustments: Sullivan GM, Feinn R. Using Effect Size-or Why the P Value Is Not Enough. [cited 2019 Apr 8]; Available from: <http://dx.doi.org/10.4300/JGME-D-12-00156.1>

4. The following statement should be listed as a reference for Bonferroni. If you took advantage of Bonferroni, could be P 0.0125 instead of 0.125?

Answer: We have added a reference for the Bonferroni. We also have corrected the p-value to 0.0125.

Adjustments: Bland, J. M., & Altman, D. G. (1995). Multiple significance tests: the Bonferroni method. *Bmj*, 310(6973), 170.

5. The reference in No. 28 as follows could not be found on the internet.

"Data were analyzed using IBM-SPSS 21.0 (28)."

28. IBM Corp. IBM SPSS Statistics for Windows. Armonk, NY: IBM Corp.; 2012.

Answer: Although the reference could not be found on the internet, the citation was taken from the

IBM website (<https://www.ibm.com/support/pages/how-cite-ibm-spss-statistics-or-earlier-versions-spss>). Therefore, we are following the official reference format from the IBM company.

6. Results P11-12

If the TS (Thanatophobia score) in MS2 difference listed in Table 2 is post-pre, then it seems to be a negative result. Also, when the text states that the TS in MS2 is not statistically decreased in MS2, you should state a specific p-value instead of stating $p < 0.05$.

Answer: Thank you for noticing. We corrected and added the minus signal in the table's result. Also, we updated the p-value.

Adjustments:

Table 2 – SEPC and Thanatophobia score means, t-test and effect size at Medical School 1 and 2

Instruments	Pre-test	Post-test	Difference Post - Pre	t	p	CI95%	Cohen's d	Effect Size Magnitude
Medical School 1 (n=78)								
SEPC - Communication	346.50 (145.17)	573.90 (124.47)	227.39 (138.50)	14.50	<0.001	258.62, 196.17	1.64	Very large
SEPC - Patient Management	338.38 (112.56)	527.50 (113.55)	189.11 (140.30)	11.90	<0.001	220.75, 157.48	1.35	Very large
SEPC - Multidisciplinary Teamwork	361.26 (124.28)	470.35 (105.77)	109.09 (118.86)	8.11	<0.001	135.17, 81.01	0.92	Large
TS	21.54 (7.56)	16.54 (7.58)	-5.00 (5.91)	-7.06	<0.001	3.59, 6.41	0.79	Moderate
Medical School 2 (n=56)								
SEPC - Communication	296.41 (147.41)	398.00 (171.64)	101.59 (151.30)	5.02	<0.001	142.11, 61.07	0.67	Moderate
SEPC - Patient Management	327.86 (133.91)	462.07 (126.55)	134.21 (132.69)	7.57	<0.001	169.75, 98.68	1.01	Large
SEPC - Multidisciplinary Teamwork	388.50 (120.18)	454.96 (120.49)	66.46 (136.82)	3.63	=0.001	103.10, 29.83	0.49	Small
TS	22.82 (8.77)	21.20 (7.67)	-1.62 (5.45)	1.96	=0.056	-0.14, -3.07	0.26	Small

Reviewer: 2

Reviewer Name: Barnaby Hole

Institution and Country: University of Bristol

Please state any competing interests or state 'None declared': None declared

Please leave your comments for the authors below

Dear team,

Thank you for listening to and responding to completely to the reviewer comments. I wish you good fortune with this paper and your important educational work.

Comments: We appreciated so much your feedback and we are pleased we could address all the comments and suggestions properly.

VERSION 3 – REVIEW

REVIEWER	Junji Haruta Keio University, Japan
REVIEW RETURNED	10-Oct-2020
GENERAL COMMENTS	As noted by the reviewer, it has been appropriately corrected. I think this is an excellent paper that describes significant findings for the reader.